# Identifying Optimal Pitch Training Load in Elite Soccer Players

**DOI:** 10.3390/jfmk10020211

**Published:** 2025-06-04

**Authors:** Adriano Titton, Elias de França, Luís Branquinho, Luís Fernando Leite de Barros, Pedro Campos, Felipe O. Marques, Igor Phillip dos Santos Glória, Erico Chagas Caperuto, Vinicius Barroso Hirota, José E. Teixeira, Nelson Valente, Pedro Forte, Ricardo Ferraz, Ronaldo Vagner Thomatieli-Santos, Israel Teoldo

**Affiliations:** 1São Paulo Futebol Clube, São Paulo 05036-040, Brazil; atitton1@hotmail.com (A.T.); luisflbarros@hotmail.com (L.F.L.d.B.); pedrocuc@terra.com.br (P.C.); felipe.marques@saopaulofc.net (F.O.M.); igorgloria@umc.br (I.P.d.S.G.); 2Centre of Research and Studies in Soccer (NUPEF), Universidade Federal de Viçosa, Viçosa 36570-900, Brazil; israel.teoldo@ufv.br; 3Interdisciplinar Graduate Program in Health Sciences, Universidade Federal de São Paulo, Santos 75985-000, Brazil; 4Human Movement Laboratory, São Judas University, São Paulo 03166-000, Brazil; ericocaperuto@gmail.com; 5Biosciences School of Elvas, Polytechnic Institute of Portalegre, 7300-110 Portalegre, Portugal; luis_branquinho@outlook.pt (L.B.); nelsonvalente@ipportalegre.pt (N.V.); pedromiguel.forte@iscedouro.pt (P.F.); 6Life Quality Research Center (LQRC-CIEQV), 2040-413 Santarém, Portugal; ricardompferraz@gmail.com; 7CI-ISCE—ISCE Douro, 4560-547 Penafiel, Portugal; 8Research Center in Sports Sciences Health Sciences and Human Development, 6201-001 Covilhã, Portugal; 9Technological Graduation in Sports and Leisure Management, FATEC of Sports, São Paulo 05818-270, Brazil; vbhirota@gmail.com; 10Department of Sports Sciences, Polytechnic Institute of Bragança, 5300-252 Bragança, Portugal; jose.eduardo@ipg.pt; 11Department of Sports Sciences, Polytechnic Institute of Guarda, 6300-559 Guarda, Portugal; 12SPRINT—Sport Physical Activity and Health Research and Innovation Center, 2040-413 Rio Maio, Portugal; 13LiveWell—Research Centre for Active Living and Wellbeing, 5300-252 Bragança, Portugal; 14Sports Department, Higher Institute of Educational Sciences of the Douro, 4560-708 Penafiel, Portugal; 15Department of Sports Sciences, University of Beira Interior, 6201-001 Covilhã, Portugal

**Keywords:** machine learning, elite soccer player, match physical performance, training load

## Abstract

Background: There are no data in the literature regarding the optimal pitch training load (PTL) for elite soccer teams during congested seasons. Objectives: This study had three goals: (1) identify whether there is an adaptation in match physical performance (MPP) in response to PTL throughout a congested season in elite soccer players; (2) identify whether MPP adaptation is specific to the coach’s PTL philosophy; and (3) identify the optimal PTL for MPP during a congested season. Method: Over two seasons, we collected data from 11,658 PTL sessions and 3068 MPP data from 54 elite male soccer players. The PTL sessions were clustered in weekly training blocks and paired with MPP for statistical and machine learning analysis. Results: Over the season, MPP increased in the mid-season and this trend decreased during the end-season. Also, MPP reflected the coach’s PTL philosophy. Further, using a machine learning (k-means) approach, we identified three different PTLs (and classified them as low-, medium-, and high-load PTL blocks). The high-load PTL block was associated with a higher MPP, while the lower PTL was associated with a lower MPP. Conclusions: PTL is closely related to MPP, and this change also reflects the coach’s PTL philosophy. Here, we report an optimal PTL that could be useful for soccer teams playing a congested season.

## 1. Introduction

Elite soccer teams can currently play more than 70 games per season. This number of matches is considered to rate the season as congested [1]. At the moment, in the first division of soccer teams, a crowded season is characterized by matches every 4 days (average per player), and a player rotation system is even implemented to mitigate and ensure more recovery time between games [2]. The literature reports that residual fatigue accompanied by muscle morphological changes (e.g., muscle fiber misalignment and muscle swelling) can last more than 72 h after a match [3,4]. In this sense, the biggest challenge for coaches and physical trainers is knowing how to implement a training load in a congested season to avoid a drop in performance and not induce poor adaptation (and, consequently, injuries).

The pitch training load (PTL) is vital in induced adaptations in athletes’ performance [5]. Little information [1,2] exists about the manipulation of the PTL in congested seasons. Previous studies have focused on analyzing the external load profile in busy seasons [1]; however, the evaluation of pitch training load and its relationship with performance still remains open. Also, the most recent studies are still investigating the impact of the external load on the internal load of young players, even in non-congested seasons [6,7]. Therefore, determining the optimal PTL and the number of training sessions between matches over the congested season is an open question in soccer. It would still be of great value to provide information about the level of different components of the optimal PTL between matches (e.g., intensity, volume, and frequency of training between games). For example, in a congested season, the frequency of weekly physical conditioning training between games has been limited to ~2 sessions (ranging from 1 to 5 sessions) depending on training opportunities [2]. However, this training block appears to be insufficient, as it has been identified that the number of training sessions and the pitch training load between matches positively correlate with several running performance variables [2]. Therefore, practical information regarding the pitch training load on adult soccer players is missing in the literature. This information is of great practical importance for coaches and still needs to be established.

In Brazilian soccer, teams with a busy calendar train their technical and tactical skills on the soccer field as part of the athletes’ physical conditioning (known as pitch training). This PTL coaching tactic characterizes the team’s playing style and running behavior. Thus, we hypothesize that PTL is reflected in match physical performance (MPP). However, to our knowledge, studies have yet to investigate the impact of the PTL practiced by coaches and staff on MPP. In this sense, examining to what extent the specificity of the PTL applied by different training philosophies affects MPP can provide important information about ideal overloads to induce adaptations in matches [8].

The literature on MPP analysis in congested seasons should be more extensive and consistent. At the same time, a review reported no changes [1], and a more recent review [9] identified a worsening or no change in running performance in elite players in periods considered congested. However, most studies on this topic have been limited to evaluating a few games (three to eight matches) [1,10]. A long-term study evaluated 52 matches over two seasons, but the team trained 5 to 6 times a week, and only home games were analyzed [11]. In another study, Dupont et al. [11] compared periods of congestion (two matches per week) with periods of one match per week and did not identify changes in running performance but identified an increase in injury rate. On the other hand, Penedo-Jamardo et al. [12] accompanied 4496 professional players for a season and identified a performance decrease in fixture periods (<4 days between matches) during the mid-season, mainly in the fullback and midfield player positions. However, these studies do not report an analysis of contextual factors, such as the training load (a vital component to induce performance adaptation). In this sense, studies assessing congestion seasons (where training opportunities between matches are scarce) and considering both home and away games are still necessary.

Considering a congested scenario, our study is guided by three fundamental hypotheses. The first is that variations in MPP over time are directly linked to PTL. The second hypothesis posits that PTL, as determined by coaches, significantly influences athletes’ MPP. Therefore, if PTL is associated with MPP and coach philosophy, then, by using a machine learning approach, it will be possible to identify the optimal PTL to induce an increase in MPP.

With these hypotheses in mind, our study aims to achieve three main objectives:Identify whether the changes in MPP and PTL throughout a congested season in elite soccer players are similar;Identify whether MPP adaptation is specific to the coach’s training load philosophy;Identify an optimal weekly PTL on MPP during a congested season.

## 2. Materials and Methods

### 2.1. Participants and Sample

Match physical performance (N = 3068 cases) and pitch training load session data (N = 11,658 cases) were collected from 54 male professional soccer players (age, 24.3 ± 4.9 years; body weight, 75.2 ± 5.7 kg; height, 178.8 ± 4.5 cm) belonging to the Brazilian First Division team during the 2022 and 2023 season. Data corresponding to 148 official matches from the two seasons were analyzed (season 2022 = 77 matches; season 2023 = 71 matches). The season of 2022 (starting with the first official match) started in January and ended in November, and there were no breaks during this entire period (that is, they had one or two matches every week). The season of 2023 (starting with the first official match) started in January and ended in December. Only data from players who complete full matches were included in the analysis. In this sense, the sample was limited to a 721 match physical performance (MPP)-case data set and their prior pitch training load (PLT) data set (see Figure 1). Thus, as presented in Figure 1, the match physical performance and pitch training load resulted in 721 paired performance cases (i.e., these data were separate counts reported jointly).

Players were classified into four positions: striker (197 cases of match performance and respective pitch training load), fullback (153 cases), winger (152 cases), and midfield (148 cases). Goalkeepers were excluded from this analysis due to the different nature of their movement patterns. This study was approved by the São Judas Tadeu University Ethics Committee (number 6.507.950; date 11 November 2023). This is a retrospective study that assessed the data from two seasons of an elite team in the first division of Brazilian football, and it complied with the ethical standards of the university and followed the guidelines of the Declaration of Helsinki.

### 2.2. Data Collection

A catapult system (Vector 7, Catapult Sports, Melbourne, Australia) with global and local positioning system devices (GPS, GLONASS & SBAS 18 Hz; LPS, Catapult ClearSky 10 Hz) combined with inertial sensors such as an accelerometer (3D +/− 16 G; sampled at 1 kHz, provided at 100 Hz), gyroscope (3D 2000 degrees/second @ 100 Hz), and magnetometer (3D ± 4900 µT @100 Hz) was used to collect data from all matches and training sessions. All three inertial sensors collected data on acceleration, force, rotation, and body orientation. The Inertial Movement Analysis (IMA) method was used to assess explosive efforts such as jumps (>40 cm), acceleration (−45 to 0, 0 to 45 degrees), deceleration (135 to 180, −180 to −135 degrees), and change of direction (COD) to the left (−135 to −45 degrees) or to the right (45 to 135 degrees). In addition, the total explosive effort (the sum of the jump, acceleration, deceleration, and COD) was recorded as the IMA explosive effort. The intensity threshold of each IMA event was set when the action occurred at >2 m/s^2^, thus characterizing an explosive action. The IMA acceleration and deceleration were set at >3 m/s^2^. The running distance producing metabolic power (W·kg^−1^) was also collected at different intensities (>20 and >55 W·kg^−1^). The player load was collected as the sum of the accelerations (>m/s^2^) of the tri-axial accelerometer. GPS methods were used to collect the total distance (m), relative distance (m/min), and running distance > 20 km/h (m), >25 km/h (m), and >30 km/h. In addition, the number of sprints (running > 25 km/h) and the maximum speed (km/h) achieved during the match or training session were recorded.

The catapult system shows suitable validity and reliability for measuring speed, acceleration, deceleration [13], jump [14], and change of direction [15]. As a previous study [13] identified a small coefficient of variation (0.9 to 1.1%) in intra- and inter-unit reliability, the players used the same device over the season.

### 2.3. Pitch Training Load Data Collection

Data on the conditioning sessions between games were collected. Pre-season training was excluded from the analysis, and a total of 11,658 training sessions were observed. Only the training loads corresponding to complete matches were included. The average number of training sessions per week before the match was calculated (reported here as a weekly training block). Players who did not complete a conditioning session but had played just one match were excluded from the analysis (see detailed description in Figure 1). Strength training and recovery sessions were not included in the analysis.

### 2.4. Contextual Factors

The team had two coaches throughout the two seasons (coach 1 = 452 cases; coach 2 = 269 cases) and used the following player positions: striker (coach 1 = 100 cases; coach 2 = 97 cases), fullback (coach 1 = 123 cases; coach 2 = 30 cases), winger (coach 1 = 80 cases; coach 2 = 72 cases), and midfield (coach 1 = 149 cases; coach 2 = 70 cases).

Coach 1 used a daily training sequence that consisted of the following activities: a first part of the training (10%) for warm-up (coordination, agility, speed, or strength), followed by exercises using small-sized games without the use of goalkeepers (15%), a third part comprising the main exercise (50%) through medium-sized games with goalkeepers, and in the final part (15%), activities for general and specific technical improvement (passes/crosses/shots). The coach’s guiding principles were the individual and collective development of the football players in technical and tactical aspects. Coach 2 followed a routine with the following characteristics: a first part (15%) for warm-up (coordination, agility, or speed), followed by exercises using small- and medium-sized games without the use of goalkeepers (25%) and the main activity (60%) with medium-/large-sized games with goalkeepers. The coach’s guiding philosophy was based on collective tactical and strategic aspects.

The home–away matches for coach 1 were home = 187, away = 229; for coach 2, they were home = 71, away = 83.

The Mann–Whitney test was used to verify if training or match frequency was similar between coaches. The match frequency was different between coaches (coach 1 = 4.59 ± 2.2 days; coach 2 = 5.3 ± 2.5 days; *p* < 0.001), but that of training sessions between matches was not (coach 1 = 2.9 ± 3.2 sessions; coach 2 = 2.9 ± 3.8 sessions; *p* = 0.67), and nor was the number of weekly training blocks (coach 1 = 2.3 ± 1.1 sessions; trainer 2 = 2.2 ± 1.0 sessions; *p* = 0.62). We quantified the number of training sessions between matches because in some cases, the players were prevented from playing as a way to preserve themselves for more important matches, and thus were only accumulating training sessions to recover. Also, we reported the amount of weekly training blocks as the weekly training sessions before each match.

To compare performance across the seasons, we divided the two seasons into quartiles, with three months for each quartile. Each quartile encompassed data from two seasons. For instance, the first quartile contained data from January, February, and March of both seasons. The data set was from 148 official matches that produced 721 player performance observations, which were divided as follows: 1st (180 of both match physical performance and session training case observations), 2nd (221 cases), 3rd (203 cases), and 4th (117 cases). Note that the numbers of cases per quartile are different; this is because there is naturally a greater number of matches in quartiles 2 and 3 (mid-season): 1st (31 matches), 2nd (46 matches), 3rd (47 matches), and 4th (24 matches). This nature in relation to the distribution of the data (imbalance over time) was taken into account in the statistical analysis using the mixed linear model (see details in the statistical analysis) [16].

### 2.5. Statistical Analysis

The data are presented as mean and standard deviation (±) or, when indicated, with a 95% confidence interval (CI). The mixed linear model was used for dependent variables to determine the difference in training load and match physical performance between the coaches and across the season. Active coach and season quartile were used as fixed effects to compare match physical performance and training load. Four contextual factors were used as covariables to compare the match physical performance between coaches: player age, player position, match avenue, and match frequency. Five contextual factors were used as covariables to compare performance across the seasons: season (2022 and 2023), player position, match avenue, player age, and coach. Because data from the same player were used multiple times, players were used as random effects (intercept model) in the mixed linear model. To examine the association between player running performance and training session load, all match data and training load data were standardized to the z-score within each player’s position. First, we performed a hierarchical cluster to generate a dendrogram. Then, based on the visual inspection of the dendrogram created from the hierarchical cluster approach, we identified that three clusters represented the best solution to solve the identified problem (is training load associated with its respective MPP?). Also, the best silhouette scores were 0.4, 0.4, and 0.3 for 2, 3, and 4 clusters, respectively, confirming objectively that 3-cluster data set separation is the best solution. Finally, we clustered the training load based on player position using a k-means approach. The k-means algorithm used Euclidean distance to compute distance and defined 100 interactions to compute the cluster centroids. The three clusters generated showed three distinct external loads from the pitch training load. The reproducibility of the clusters created in this study was tested in the database itself. To complete this, clustering was repeated several times from different classification orders of the database (i.e., changing the initial k centers). As a result, the k-means converged to the same cluster profile regardless of the initial k center, thus demonstrating that our database is sufficiently large and representative of elite soccer players. To validate the clusters of training load (i.e., to verify if clusters were different between them), we use one-way ANOVA (using clusters as factors and the z-score of training load variables as dependent variables) followed by the Duncan post hoc test. The difference in match physical performance in the function of training load was verified with the mixed linear model using coach and training load cluster as fixed factors; player position, coach, season, match avenues, and match frequency as covariables; and player as a random effect. The η2 effect size was reported as ≥0.01 as small, ≥0.06 as medium, and ≥0.14 as a large effect size. All analyses were performed using the statistical package IBM SPSS Statistics v.26.0.

## 3. Results

### 3.1. Changes in Match Physical Performance and Pitch Training Load over the Season

Figure 2 presents the changes in performance throughout a congested season. In general, MPP increased at the beginning of the season and then returned to baseline or decreased at the end of the season. Specifically, throughout the season, a significant improvement (all *p* < 0.05) in in-game performance was identified for total and relative distance, distance > 20 km/h, distances > 20 and 55 W·kg^−1^, explosive efforts, player load, chance of direction (COD), and acceleration and deceleration (<2 m/s^2^). The in-game performance returned to baseline at the end of the season for all variables except deceleration and distance > 55 w, which further decreased. There was a decrease (all *p* > 0.05) in in-game performance throughout the season for maximum speed, distances > 25 and >30 km/h, and acceleration/deceleration (>3 m). The effect size from the MPP variation over the season was small on all variables (all ≤ 0.03).

In general, the training load started higher at the beginning of the season and tended to decrease or remain stable over the season. Specifically, when comparing the last quartile to the first quartile of the season, there was a significant decrease (all *p* < 0.05) in the training load for total distance, distances > 20 and 25 km/h, distances > 20 and 55 w, maximum speed, number of sprints, explosive effort, player load, accelerations/decelerations, and COD (Figure 3). There was no change (all *p* > 0.05) in the training load throughout the season for relative distance and a distance of 30 km/h (Figure 3). The effect size from the PTL variation over the season was small on all variables (all ≤ 0.03).

Figure 4 shows that PTL size decreases over the season. Specifically, there was a lower amount of training between matches (depicted in Figure 4 as training sessions between matches) and before games (depicted in Figure 4 as a weekly training block) in the second and third trimester. Also, the days between games (depicted in Figure 4 as match frequency) decreased in the third semesters, indicating that players had fewer days to recover and train during this period. This revealed a connected variation throughout the season per player in match frequency, training between matches, and weekly training blocks.

### 3.2. Changes in Match Physical Performance According to Coach Training Load

Table 1 shows that match physical performance is related to a higher training load of each respective variable. For example, when compared to coach 1, coach 2 has a higher match physical performance in the variables of total distance, explosive efforts, COD, and deceleration, and this difference is reflected in their weekly training. On the other hand, coach 1 has a higher match physical performance in the straight-line running variables (i.e., relative distance and distance > 20 W·kg^−1^) and a lower RHIE block recovery time when compared to coach 2. At the same time, the same variables have a greater load during the weekly training blocks.

### 3.3. Identifying Optimal Pitch Training Load Associated with Match Physical Performance

To answer the questions of whether the training load per se and coaches’ related training load may affect match physical performance, we used a machine learning (k-means) approach to identify different training loads. Thus, Table 2 presents three training load clusters, namely low-load (with 145 training load cases), mid-load (with 344 training load cases), and high-load (with 208 training load cases). The three training clusters are different from each other, except for distance > 30 km/h (m) and RHIE block recovery time. Also, match frequency, the training block before the matches, and training between matches are different between the three clusters. This indicates that a training load is applied inside these technical features.

Next, in Figure 5, we plot the match physical performance as a function of the training load described in Table 2. In the high-load cluster, there is a higher match performance associated with several variables when compared with the mid- and lower-load clusters. However, there is no significant dose response for the very high-intensity variables (i.e., distance > 30 km/h, maximum speed, and accelerations/decelerations > 3 m). The effect size from the training load clusters is negligible on all variables (all ≤ 0.01).

## 4. Discussion

The main finding of this study was the identification that match physical performance reflects the pitch training load in a specific way. Specifically, due to our massive number of weekly training blocks (721 blocks) collected over two seasons, we could identify three distinct training loads (i.e., low, moderate, and high training load blocks). Interestingly, the high training load identified in this study was positively associated with the athletes’ performance. Furthermore, the training load and match physical performance data allowed us to identify a specific pattern for the two coaches, which allowed us to recognize that the high-load pitch training throughout the seasons of the two coaches was associated with better running performance on the field in a specific way. To our knowledge, in addition to the data presented in a previous study by our group [2], there are no other data in the literature that provide a guide to the pitch training load in teams that are involved in congested seasons (over 70 games per season). Therefore, based on our data, coaches and physical trainers can use the methodology described in this study to identify and prescribe the optimal training load throughout the season for soccer players.

Another important finding in our study was the identification of changes in performance throughout the season. Specifically, we identified an improvement in match physical performance during the 2nd and 3rd quartiles, followed by a performance return to baseline (i.e., like to the 1st quartile) during the 4th quartile. We hypothesize that this decrease in training load is a consequence of a lack of player readiness, mainly due to progressive fatigue over the season [6,7,8]. Thus, a strategy to monitor fatigue on a daily basis is necessary to monitor negative changes in performance over the season [17]. For this, the machine learning analysis is a promissory approach due to it being non-invasive and non-time-demanding (for athletes and coaches) [8].

We also identified a decrease in training load throughout the season in several variables. This decrease occurred mainly in variables related to explosive effort and peak velocity, suggesting that very high-intensity variables are affected at the end of the season. However, such a decrease in training load did not negatively impact match physical performance. The average reduction in training load over the season, in general, was from the high-load to mid-load blocks. Thus, none of the variables decreased to a low load. As mentioned for match physical performance, a hypothesis for this decrease could also be related to a lack of player readiness [6,7,8]. Another hypothesis is that a lack of time imposes this decrease in training load (see Table 2). For instance, the lower training load occurs concomitantly over the season with reduced days between matches and training blocks. Thus, it is suggested that a decrease in training load is necessary to avoid overtraining.

This study also showed that high-load pitch training was significantly associated with match physical performance. Low- and mid-load pitch training was associated with low and medium running performance in matches, respectively (see Figure 5). Thus, a change in match physical performance throughout the season may be related to a change in pitch training load over the season. However, caution is needed to interpret these data; as presented in Table 2, the training load was also associated with the length of training block sessions and with match frequency. For example, low-load pitch training was associated with a high match frequency (every three days) and ~1.6 session training blocks. In comparison, high-load pitch training was associated with a lower match frequency (every 4.5 days) and ~2.6 session training blocks. Thus, we cannot discard that low match performance could be a consequence of fatigue between matches, and that low load could be a consequence of the downward modulation of training load to avoid overtraining the soccer players during over-congested matches (i.e., every three days).

The differences in match physical performance corresponds to the same differences in pitch training load across the three clusters (low, medium, and high load). In addition to physical conditioning, pitch training is used for technical and tactical purposes, which can also impact the field movement profile because tactical/technical training aims to create movement behavior (i.e., playing style). However, it is undeniable that pitch training load impacts performance negatively or positively [5]. For instance, in a study by Guerrero-Calderón et al. [5], a negative load was identified for the total distance in training sessions (negatively affecting total distance and high-intensity runs in the matches); on the other hand, a positive training load was identified for high-intensity running activity (>25 km/h). Our data did not identify a negative effect but a positive one for both variables. In our previous study [2], where we used data from one season (77 matches and from just a single coach), we identified that a high-impact activity training load (such as explosive effort, COD, jump, and deceleration) or running in a straight line had cross-interference on match physical performance. For example, when blocks of impact training (such as COD, accelerations, decelerations, and jumps) are performed, they cause a negative impact on straight-line running variables (such as running at speeds > 20, >25, or >30 km/h and total running distance) or vice versa. Our results suggest that this cross-interference may occur because differences in field performance reflect the training load in the three clusters (low, medium, and high load). For instance, COD, explosive efforts, and acceleration/deceleration activities are emphasized in small-sized games during pitch training load (generates a greater volume and intensity for these activities), while straight-line running activities are emphasized in all-field pitch training. This different approach will be reflected in match physical running performance during the official games. In this sense, trainers should periodize these loads to avoid cross-interferences and to induce desirable adaptation [18].

Our description of training loads provides information on optimal pitch training load prescription. For instance, a previous study by our group [2] and our current data also demonstrate that training volume (training between matches) has a positive impact on straight-line running performance (described in Figure 2). Also, accumulating pitch training between matches (i.e., volume) positively affects players’ performance in a congested season. In addition, our data suggest that a 4.5-day match frequency may be sufficient to induce better performance than a 3-day match frequency. A previous study assessing only home matches indicated that a short period of match congestion (2–4-day match frequency) did not negatively impact running performance [1]. Our data assessed both home and away matches, thus adding greater ecological validity to the analysis. Therefore, the sum of our data suggests that one or two matches per week (i.e., every 4.5 days) does not negatively impact match physical performance [2,11]. However, some players have a match frequency of >7 days, and this remains an open question. Thus, the ideal match frequency remains an open question in match physical performance. Albeit, a high match frequency (i.e., two matches per week) did not negatively affect match physical performance but increased injury risk [1]; thus, coaches should be cautious during congested periods, and adopting rotation between players is advised. This caution is crucial for maintaining player health and performance. A previous study [19] identified that chronic high-load training could reduce the risk of injury incidence when compared to chronic low-load training. In this sense, future studies should investigate the relationship between training load and injury incidence in soccer during the congestion season.

It is important to note that here we conducted a retrospective observational study, and thus the significant association between pitch training load and match physical performance may not be cause–effect. In this sense, future studies with experimental designs (randomized controlled approach, observing athletes with different pitch training loads) aiming to identify a dose–response relationship are still necessary.

## 5. Conclusions

Match physical performance changes throughout a congested season, and these changes closely reflect the pitch training load. Match physical performance reflects the coach’s pitch training load. The machine learning approach is an efficient and practical tool to identify the optimal training load associated with a higher match physical performance. Here, we report a beneficial pitch training load that could be useful in teams playing two matches per week. Future studies in other scenarios (non-congested seasons) are needed to confirm our observational findings.

## Figures and Tables

**Figure 1 jfmk-10-00211-f001:**
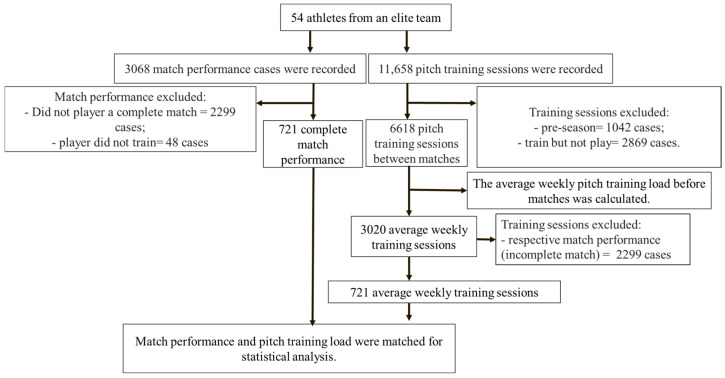
Flowchart with inclusion and exclusion criteria for match physical performance and pitch training data.

**Figure 2 jfmk-10-00211-f002:**
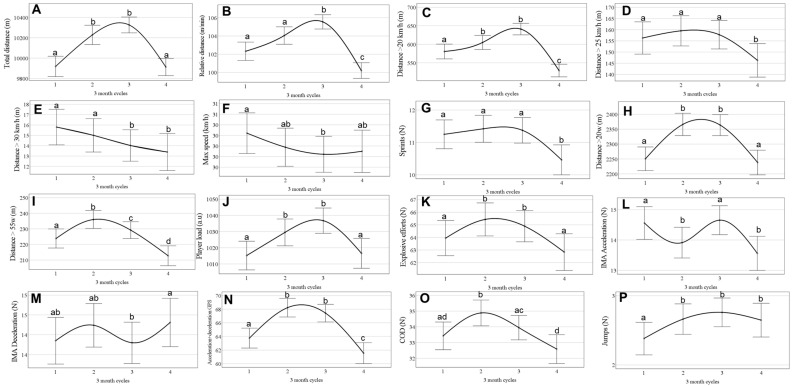
Change in match physical performance over season, divided into three-month quartiles (data are mean estimated values and 95% CI). (**A**) Total distance (m); (**B**) relative distance; (**C**) distance > 20 km/h; (**D**) distance > 25 km/h; (**E**) distance > 30 km/h; (**F**) maximum speed; (**G**) number of sprints; (**H**) distance > 20 W·kg^−1^; (**I**) distance > 55 W·kg^−1^; (**J**) player load; (**K**) explosive effort; (**L**) IMA acceleration; (**M**) IMA deceleration; (**N**) GPS acceleration + deceleration; (**O**) change of direction; (**P**) jumps. Different letters denote statistical difference (*p* < 0.05) between quartiles.

**Figure 3 jfmk-10-00211-f003:**
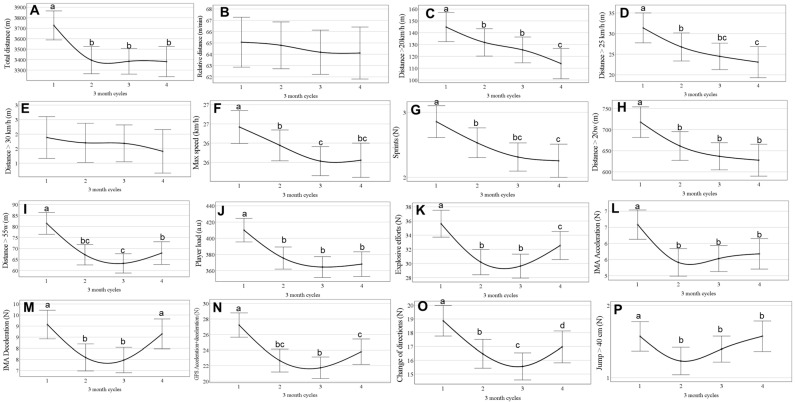
Change in weekly training load over season, divided into three-month quartiles (data are mean estimated values and 95% CI). (**A**) Total distance (m); (**B**) relative distance; (**C**) distance > 20 km/h; (**D**) distance > 25 km/h; (**E**) distance > 30 km/h; (**F**) maximum speed; (**G**) number of sprints; (**H**) distance > 20 W·kg^−1^; (**I**) distance > 55 W·kg^−1^; (**J**) player load; (**K**) explosive effort; (**L**) IMA acceleration; (**M**) IMA deceleration; (**N**) GPS acceleration + deceleration; (**O**) change of direction; (**P**) jumps. Different letters denote statistical difference (*p* < 0.05) between quartiles.

**Figure 4 jfmk-10-00211-f004:**
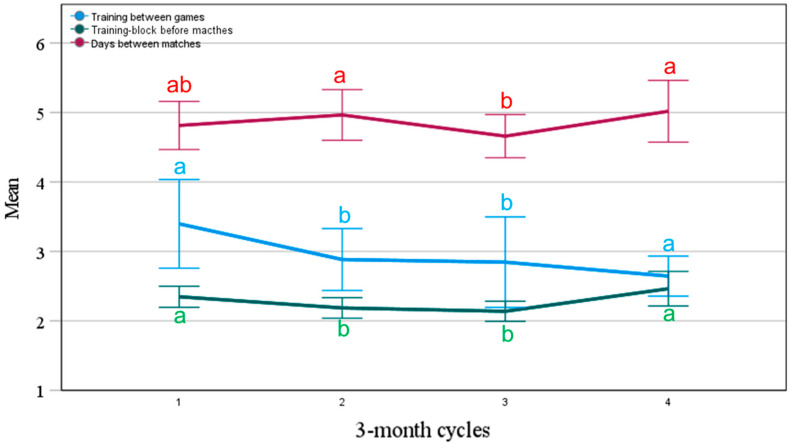
Changes in training and match frequencies over the season, divided into three-month quartiles (data are mean predicted values and 95% CI). We used Kruskal–Wallis followed by Bonferroni post hoc to verify the difference between quartiles. Different letters denote *p*< 0.05 across the quartiles.

**Figure 5 jfmk-10-00211-f005:**
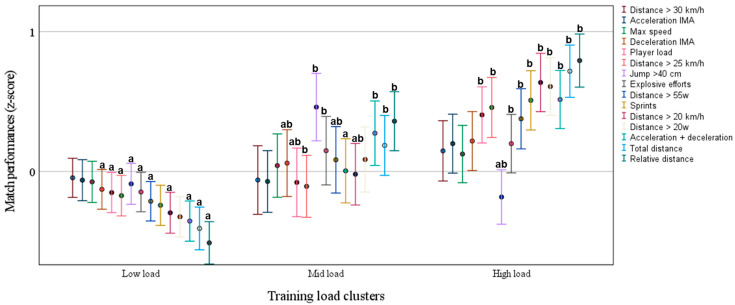
Match physical performance variables as a function of training load (data are mean estimated values and 95% CI). Different letters indicate statistical differences at *p* ≤ 0.5 when comparing the training clusters.

**Table 1 jfmk-10-00211-t001:** Description [mean and (95% CI)] and comparison of match physical performance and week training load under two coaches.

Variables	Match Physical Performance		*p* Value	Mean Week Training Load		*p* Value
	Coach 1 (N = 452)	Coach 2 (N = 269)		Coach 1 (N = 452)	Coach 2 (N = 269)	
Total distance (m)	11,262.1 (11,021.8/11,502.4)	11,482.9 (11,237.4/11,728.4)	<0.001	3060.2 (2940.6/3179.7)	3502.9 (3373.9/3631.9)	<0.001
Relative distance (m/min)	98.7 (96.4/100.9)	92.4 (90.1/94.7)	<0.001	62.6 (61.5/63.8)	56.5 (54.9/58.0)	<0.001
Distance >20 km/h (m)	602.3 (540.7/670.9)	608.8 (546.7/670.9)	0.624	126.1 (111.2/142.6)	103.2 (85.9/120.4)	<0.001
Distance >25 km/h (m)	152.4 (127.6/177.2)	163.2 (138.1/188.3)	0.065	28.9 (22.5/35.4)	22.1 (15.3/28.9)	<0.001
Distance >30 km/h (m)	14.1 (9.6/18.5)	13.5 (8.9/18.1)	0.672	2.1 (0.9/3.1)	2.0 (0.8/3.1)	0.896
Sprints (N)	10.9 (9.2/12.7)	11.3 (9.5/13.0)	0.441	2.3 (1.8/2.7)	1.7 (1.2/2.2)	<0.001
Max. speed (km/h)	30.3 (29.9/30.7)	30.1 (29.7/30.6)	0.540	26.2 (25.8/26.6)	24.2 (24.2/25.1)	<0.001
Accelerations (N)	17.4 (15.5/19.3)	17.1 (15.1/19.0)	0.609	4.9 (4.6/5.2)	5.6 (5.2/6.1)	0.001
Decelerations (N)	18.4 (16.7/20.1)	22.3 (20.5/24.1)	<0.001	7.1 (6.5/7.5)	8.6 (7.9/9.2)	<0.001
Jumps > 40 cm (N)	3.6 (3.3/4.4)	4.3 (3.6/5.0)	0.135	0.9 (0.7/1.1)	2.1 (1.9/2.3)	<0.001
Explosive effort	78.4 (73.3/83.4)	86.8 (81.6/92.0)	<0.001	27.5 (25.8/29.1)	32.1 (30.1/34.2)	<0.001
COD (N)	40.9 (37.3/44.6)	45.8 (42.1/49.5)	0.011	14.9 (13.8/16.1)	17.1 (15.6/18.4)	0.003
Distance > 20 W·kg^−1^ (m)	2384 (2260.8/2508.3)	2287.5 (2161.3/2413.7)	0.002	623.9 (586.8/661.1)	560.6 (517.4/603.9)	0.001
Distance > 55 W·kg^−1^ (m)	243. (215.7/270.6)	246.4 (218.8/274.0)	0.555	71.2 (62.9/79.4)	65.99 (57.1/74.9)	0.109
Player load (u.a)	1147 (1105.7/1189.9)	1157 (1114.8/1200.0)	0.302	337.3 (323.7/350.9)	366.3 (347.5/379.8)	<0.001

Data are mean (95% CI) estimated values.

**Table 2 jfmk-10-00211-t002:** Clusters of training load cases. Data are mean values and 95% CI.

Variables	Cluster Training	Mean	95% CI		Cluster Comparisons (Different Letters Denote Statistical Difference)
			LB	UB	
Match frequency (days)	High load	4.5	4.1	4.8	A
	Mid load	3.9	3.6	4.1	B
	Low load	2.9	2.8	2.9	C
Training block before matches (N)	High load	2.6	2.4	2.7	A
	Mid load	2.4	2.3	2.5	B
	Low load	1.6	1.5	1.7	C
Training between games (N)	High load	4.1	3.5	4.8	A
	Mid load	2.9	2.6	3.3	B
	Low load	1.6	1.5	1.7	C
Total distance (m)	High load	4129.3	4071.5	4187.0	A
	Mid load	3163.5	3131.6	3195.5	B
	Low load	1910.3	1821.5	1999.2	C
Relative distance (m/min)	High load	68.2	66.5	69.9	A
	Mid load	59.6	58.6	60.6	B
	Low load	51.5	49.0	54.0	C
Distance > 20 km/h (m)	High load	155.1	143.6	166.6	A
	Mid load	92.6	86.0	99.1	B
	Low load	47.4	41.1	53.8	C
Distance > 25 km/h (m)	High load	32.4	28.1	36.7	A
	Mid load	18.3	16.3	20.4	B
	Low load	9.7	7.8	11.7	C
Distance >3 0 km/h (m)	High load	2.2	1.3	3.0	A
	Mid load	1.0	0.7	1.3	B
	Low load	0.4	0.0	0.8	B
Explosive effort (N)	High load	37.3	35.5	39.0	A
	Mid load	28.9	27.6	30.1	B
	Low load	16.8	15.2	18.3	C
Maximum speed (km/h)	High load	27.0	26.5	27.4	A
	Mid load	25.4	25.1	25.7	B
	Low load	23.8	23.3	24.3	C
Distance > 55 W·kg^−1^ (m)	High load	88.9	83.4	94.4	A
	Mid load	61.1	57.8	64.3	B
	Low load	29.8	26.6	33.0	C
Player load(N)	High load	432.3	424.4	440.2	A
	Mid load	336.7	332.1	341.3	B
	Low load	209.5	199.7	219.3	C
Sprints (N)	High load	2.5	2.2	2.8	A
	Mid load	1.5	1.3	1.6	B
	Low load	0.8	0.6	0.9	C
GPS Acceleration + deceleration (N)	High load	29.2	27.6	30.8	A
	Mid load	20.9	19.9	21.8	B
	Low load	10.1	9.0	11.1	C
IMA Acceleration (N)	High load	6.4	6.1	68	A
	Mid load	5.2	5.0	5.5	B
	Low load	3.2	2.9	3.6	C
IMA Deceleration (N)	High load	9.6	9.0	10.3	A
	Mid load	7.6	7.2	8.0	B
	Low load	4.2	3.7	4.7	C
Change of direction (N)	High load	20.4	19.3	21.4	A
	Mid load	15.3	14.5	16.0	B
	Low load	8.9	8.0	9.7	C
Jump > 40 cm	High load	0.68	0.81	1.12	A
	Mid load	1,60	0.58	0.81	B
	Low load	0.78	0.34	0.64	A
Distance >20 W·kg^−1^ (m)	High load	812.1	787.0	837.1	A
	Mid load	547.8	533.3	562.2	B
	Low load	298.6	277.4	319.8	C

In the “Cluster Comparisons” column, different letters indicate statistical differences at *p* ≤ 0.5 when comparing the training clusters. Captions: LB, lower bound; UB, upper bound.

## Data Availability

The raw data supporting the conclusions of this article will be made available by the authors on request.

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
