# Peer review of "Identifying Optimal Pitch Training Load in Elite Soccer Players"

_jfmk, 2025, doi:10.3390/jfmk10020211_

Round 1
Reviewer 1 Report
Comments and Suggestions for Authors
Many thanks to the Editor for the opportunity to revise this exceptional article entitled “Identifying Optimal Pitch Training Load in Elite Soccer Players”, in which the Authors have collected a very huge amount of pitch training load sessions in professional soccer players, and operated a very sophisticated statistical analysis to support their aims and objective. I have no methodological requests, just few grammars or small suggestions to improve this practical work.
May I suggest inserting a very short background in the abstract? It may serve to introduce the objectives of the work.
Can you rephrase the following sentence? Lines 43-44 “Elite soccer teams can currently play more than 70 games per season; a higher number was detected earlier (i.e., 50 games).” Did you mean that years ago matches were only around 50, but recently they increased up to around 70?. It could be hard to understand.
What did you mean with “morphological changes” in line 48? Can you specify with some examples as this is a very generic term?
Can you solve the grammar error in line 192? I think it is meant “season” and “return to baseline”. And in line 209 “star higher “. And line 211 “first of the season”, did you mean “first season?”
Can you explicit the terms RHIE and COD also in the first time of appearance in the main text? (around lines 195-196).
Can you increase the size of the X and Y axys in Figures 2 and 3? They could be very difficult to read.
Author Response
Thanks to reviewer two for the important points to improve the paper. We accept all the reviewer's comments as important to improve the article. Below we comment on what was done in each point.
Point: May I suggest inserting a very short background in the abstract? It may serve to introduce the objectives of the work.
Response: We add a short background.
Point: Can you rephrase the following sentence? Lines 43-44 “Elite soccer teams can currently play more than 70 games per season; a higher number was detected earlier (i.e., 50 games).” Did you mean that years ago matches were only around 50, but recently they increased up to around 70?. It could be hard to understand.
Response: We delete “a higher number was detected earlier (i.e., 50 games)”. This only highlights the fact that elite teams currently play congested seasons.
Point: What did you mean with “morphological changes” in line 48? Can you specify with some examples as this is a very generic term?
Response: We made the text more specific.
Point: Can you solve the grammar error in line 192? I think it is meant “season” and “return to baseline”. And in line 209 “star higher “. And line 211 “first of the season”, did you mean “first season?”
Response: Thank you for the observations on the typos. All corrected.
Point: Can you explicit the terms RHIE and COD also in the first time of appearance in the main text? (around lines 195-196).
Response: The acronym name was added.
Point: Can you increase the size of the X and Y axys in Figures 2 and 3? They could be very difficult to read.
Response: We redid the figures by increasing the size of the letters.
Reviewer 2 Report
Comments and Suggestions for Authors
Manuscript Title: Identifying Optimal Pitch Training Load in Elite Soccer Players
Manuscript ID: jfmk-3588251
General Comments
This manuscript explores the relationship between pitch training load (PTL) and match physical performance (MPP) in elite soccer players across two congested seasons, using a large dataset and a combination of statistical and machine learning approaches. The topic is of practical relevance for high-performance practitioners and offers a novel contribution through its use of cluster analysis to identify optimal training loads.
However, certain methodological and interpretative aspects require clarification or further development to ensure the robustness and reproducibility of the findings. Below are detailed comments that aim to assist the authors in improving the clarity and scientific rigour of the manuscript.
Major Comments
- Justification and methodological detail of cluster analysis
- The manuscript does not explain how the number of clusters (k = 3) in the k-means analysis was determined. Could the authors clarify whether any formal validation approach (e.g., elbow method, silhouette score) was used? Including such validation would strengthen the credibility of the clustering approach.
- Please specify the distance metric used (e.g., Euclidean) and whether all training load variables were standardized prior to clustering. This detail is crucial, as k-means is sensitive to scale differences among variables.
- Confounding between training load and match frequency
- The authors are encouraged to adjust for match frequency in the mixed models assessing the relationship between PTL cluster and MPP. If this is not feasible, the limitation should be clearly acknowledged and discussed in more detail.
- Description of coaching “philosophies”
- The study interprets differences in PTL and MPP between coaches as indicative of distinct training “philosophies.” However, no qualitative or contextual information is provided to support this claim.
- Please provide a brief, descriptive summary of the types of training typically used by each coach (e.g., emphasis on small-sided games, conditioning drills, tactical sessions) to substantiate the claim of differing coaching philosophies.
- Clarity in season segmentation (quartiles)
- You say there are 148 games in total → so each quartile should include about 37 games. Instead, the "cases" reported per quartile (180, 221, 203, 117) are not evenly distributed. This raises two questions:
- What do these numbers represent? Are they games, training blocks, player-game observations or something else?
- How was the division actually done? Is it chronological (e.g. every 3 months)? By number of games? By number of MPP cases? Or something else?
- Please clarify whether the quartiles refer to equal calendar periods or an equal number of matches/sessions and ensure consistency between match counts and reported sample sizes.
Here we see that the numbers of “cases” reported for each quartile do not match the total number of games declared (148).
If each game corresponds to a certain number of “cases” (i.e. player-game observations or training blocks), then there should be a logical coherence between:
- Number of games per quartile.
- Number of observations analysed.
But here we see an unbalanced distribution: for example, the 4th quartile has many fewer cases (117) than the 2nd (221), even if it is not clear whether they have the same number of games or the same time interval.
- Interpretation of causal relationships
- Several statements in the manuscript suggest a causal relationship between PTL and MPP (e.g., “high training load improves performance”). Given the observational nature of the study, causality cannot be inferred.
- The authors are advised to revise the language throughout the manuscript to refer to associations rather than causal effects, and to explicitly acknowledge this limitation in the Discussion section.
- Statistical reporting consistency
- There appears to be an inconsistency in the reporting of p-values in Section 3.1: “a significant decrease (all P > 0.05)” is contradictory. Please review and correct all statistical statements for accuracy.
- Where relevant, please report effect sizes (e.g., mean differences or standardized effect sizes) in addition to p-values to support interpretation of the practical significance of findings.
Minor Comments
- Clarification of abbreviations and table labels
- In Table 2, the abbreviations “LB” and “UP” (lower and upper bounds of the confidence interval) are not defined. Please include a footnote or caption explaining these terms.
- Ensure all abbreviations (e.g., RHIE, MPP, PTL) are defined upon first use in both the Abstract and the main text.
- Typographical corrections
- Please correct the following:
- “PTL trends to decrease” → “PTL tended to decrease”
- “star higher” → “start higher”
- “pith training load” → “pitch training load”
- “tree clusters” → “three clusters”
- Figure and table presentation
- In Figures 2–4, ensure that all subplots are clearly labelled (A, B, C…) and that figure legends are self-explanatory.
- In Table 1, consider aligning the decimals and confirming whether values are presented as “mean (95% CI)” or “mean ± 95% CI”.
- Details on missing data handling
- Please clarify whether there were any missing data among training sessions or match observations, and if so, how they were handled in the analyses.
- Literature contextualization
- The Introduction states that “no studies have addressed managing PTL during a congested season in adult soccer players.” However, previous reviews have examined this topic, albeit using broader definitions of training load (e.g., Carling et al., 2015). Consider revising the claim to reflect more accurately the novelty of your study in terms of pitch-specific load.
Recommendation
I believe this study addresses an important topic in the field of applied sport science. The dataset is strong, and the analytic approach is promising. However, I recommend major revisions to clarify methods, address potential confounding factors, and moderate causal interpretations. I look forward to reviewing a revised version that addresses these concerns.
Author Response
Thanks to reviewer two for the important points to improve the paper. We accept all the reviewer's comments as important to improve the article. Below we comment on what was done in each point.
Major Comments
- Justification and methodological detail of cluster analysis
- The manuscript does not explain how the number of clusters (k = 3) in the k-means analysis was determined. Could the authors clarify whether any formal validation approach (e.g., elbow method, silhouette score) was used? Including such validation would strengthen the credibility of the clustering approach.
Response: We perform the hierarchical cluster to generate a dendrogram. Then, based on the visual inspection of the dendrogram created from the hierarchical cluster approach, we identified that three clusters represent the best solution to solve the identified problem. The three clusters generated were three distinct external loads from field training and game performance. This information was add at lines 196 to 203
- Please specify the distance metric used (e.g., Euclidean) and whether all training load variables were standardized prior to clustering. This detail is crucial, as k-means is sensitive to scale differences among variables.
Response: We add this information at lines 202 and 197.
Confounding between training load and match frequency
- The authors are encouraged to adjust for match frequency in the mixed models assessing the relationship between PTL cluster and MPP. If this is not feasible, the limitation should be clearly acknowledged and discussed in more detail.
Response: We adjust PTL cluster and MPP for match frequency (Figure 6). Also, to eliminate the effect of player position on cluster creation, z-score was calculated based on the players’ positions (within the players' positions). Also the clusters created were made within the players' positions. This removes the effect of player position on performance metrics.
- Description of coaching “philosophies”
- The study interprets differences in PTL and MPP between coaches as indicative of distinct training “philosophies.” However, no qualitative or contextual information is provided to support this claim.
- Please provide a brief, descriptive summary of the types of training typically used by each coach (e.g., emphasis on small-sided games, conditioning drills, tactical sessions) to substantiate the claim of differing coaching philosophies.
Response: we added a qualitative description in lines 170 to 181.
- Clarity in season segmentation (quartiles)
- You say there are 148 games in total → so each quartile should include about 37 games. Instead, the "cases" reported per quartile (180, 221, 203, 117) are not evenly distributed. This raises two questions:
- What do these numbers represent? Are they games, training blocks, player-game observations or something else?
- How was the division actually done? Is it chronological (e.g. every 3 months)? By number of games? By number of MPP cases? Or something else?
- Please clarify whether the quartiles refer to equal calendar periods or an equal number of matches/sessions and ensure consistency between match counts and reported sample sizes.
Here we see that the numbers of “cases” reported for each quartile do not match the total number of games declared (148).
If each game corresponds to a certain number of “cases” (i.e. player-game observations or training blocks), then there should be a logical coherence between:
- Number of games per quartile.
- Number of observations analysed.
But here we see an unbalanced distribution: for example, the 4th quartile has many fewer cases (117) than the 2nd (221), even if it is not clear whether they have the same number of games or the same time interval.
Response: We have reorganized the text to explain the distribution in lines 181-186.
- Interpretation of causal relationships
- Several statements in the manuscript suggest a causal relationship between PTL and MPP (e.g., “high training load improves performance”). Given the observational nature of the study, causality cannot be inferred.
- The authors are advised to revise the language throughout the manuscript to refer to associations rather than causal effects, and to explicitly acknowledge this limitation in the Discussion section.
Response: Thanks for the suggestion, we adapted the text on line 310, 313.
- Statistical reporting consistency
- There appears to be an inconsistency in the reporting of p-values in Section 3.1: “a significant decrease (all P > 0.05)” is contradictory. Please review and correct all statistical statements for accuracy.
Response: Thanks for the observation. Fixed.
6.1. - Where relevant, please report effect sizes (e.g., mean differences or standardized effect sizes) in addition to p-values to support interpretation of the practical significance of findings.
Response: we report the effect size for changes in MPP (Fig 2) and PTL (Fig 3) over the season. The effect size was also reported when comparing MPP between clusters (Fig 5)..
Minor Comments
- Clarification of abbreviations and table labels
- In Table 2, the abbreviations “LB” and “UP” (lower and upper bounds of the confidence interval) are not defined. Please include a footnote or caption explaining these terms.
- Ensure all abbreviations (e.g., RHIE, MPP, PTL) are defined upon first use in both the Abstract and the main text.
Response: Thanks for the pointers. Typos fixed.
- Typographical corrections
- Please correct the following:
- “PTL trends to decrease” → “PTL tended to decrease”
- “star higher” → “start higher”
- “pith training load” → “pitch training load”
- “tree clusters” → “three clusters”
Response: Thanks for the pointers. Typos fixed.
- Figure and table presentation
- In Figures 2–4, ensure that all subplots are clearly labelled (A, B, C…) and that figure legends are self-explanatory.
- In Table 1, consider aligning the decimals and confirming whether values are presented as “mean (95% CI)” or “mean ± 95% CI”.
Response: decimals values aligned and values presented as mean (95% CI)
- Details on missing data handling
- Please clarify whether there were any missing data among training sessions or match observations, and if so, how they were handled in the analyses.
Response: we discarded samples with missing values. The N samples for comparisons and cluster creation were reported throughout the paper.
- Literature contextualization
- The Introduction states that “no studies have addressed managing PTL during a congested season in adult soccer players.” However, previous reviews have examined this topic, albeit using broader definitions of training load (e.g., Carling et al., 2015). Consider revising the claim to reflect more accurately the novelty of your study in terms of pitch-specific load.
Response: Thanks for the suggestion, indeed Carling discusses training load in busy seasons. We have organized the text with the reference to the author.
Round 2
Reviewer 2 Report
Comments and Suggestions for Authors
Major Comments
- Justification and methodological detail of cluster analysis
- Thank you for the response. The use of hierarchical clustering and dendrogram inspection provides a useful initial estimate of the number of clusters. However, including a formal validation index such as the silhouette coefficient or Davies–Bouldin index would further strengthen the argument that k = 3 is the most appropriate choice for this dataset. Even a posteriori validation (e.g., silhouette score for k = 2, 3, 4) could be insightful.
- Confounding between training load and match frequency
- Thank you for the response. However, Figure 6 is not included in the current version of the manuscript. To improve transparency, we suggest either correcting this reference or clearly illustrating in the results (possibly in a table or figure) how the adjustment for match frequency influenced the relationship between PTL cluster and MPP. Additionally, a more explicit mention of this adjustment in the Methods section (e.g., Section 2.5) would strengthen the clarity of the statistical approach.
- Clarity in season segmentation (quartiles)
- Thank you for your reply and for adding explanatory content to the revised manuscript regarding the division of the dataset into quartiles. The clarification that the quartiles correspond to fixed three-month periods across both seasons helps to contextualize the temporal structure of the analysis. This was a helpful addition to the text and contributes to a better understanding of your methodological approach. That said, I would like to offer some further observations and suggestions, as a few aspects remain ambiguous or potentially confusing for readers. First, I would like to point out a small inconsistency in your response: you mentioned that the relevant clarification had been added between lines 181 and 186. However, upon reviewing the revised manuscript, the passage describing the quartile division appears in lines 193 to 200. This is a minor but important detail that should be corrected, as incorrect line references can cause unnecessary confusion during peer review and editorial processing. More substantively, although it is now stated that the numbers reported per quartile (e.g., 180, 221, 203, 117) represent “cases” and not matches, it remains unclear what exactly each “case” refers to. At various points in the manuscript, it seems that a case might correspond to a player-match observation, possibly combined with its corresponding weekly training block. However, in the revised paragraph, the wording "180 of both match physical performance and session training case observations" is ambiguous and open to interpretation. Does this mean that each case includes both a match performance observation and the associated training block? Or that these are separate counts being reported jointly? Given the centrality of these data to your clustering and mixed-model analyses, I strongly recommend clearly defining what is meant by a "case" early in the Methods section — ideally in section 2.1, where the sample is described. Explicitly stating that, for example, a case corresponds to a player who played a full official match and for whom the prior training data are available, would help avoid any confusion and enhance the transparency of your dataset construction. Furthermore, although your clarification attributes the unequal distribution of cases across quartiles to the higher number of matches typically played during the middle of the season, I believe this point could be made more explicitly and quantitatively. Reporting the number of matches per quartile or including a brief table or figure summarizing this information, would help demonstrate the internal consistency of the dataset — particularly given that 148 matches yielded 721 observations. This would also make it easier for readers to assess how evenly or unevenly the observational load was distributed across the season and how that may have influenced the balance of statistical comparisons. This is especially relevant for the fourth quartile, which contains significantly fewer observations (117) compared to the second (221), even though each period spans three months. It would be helpful to explain not just that this is a natural outcome of the competitive calendar, but also whether and how such sample imbalance was accounted for analytically — for instance, whether any adjustments or weighting were applied in your mixed models to account for the lower volume of data in some quartiles.
- Interpretation of causal relationships
- Thank you for making the suggested edits around lines 310–313. The language in that section has indeed been revised appropriately to reflect associations rather than causality. However, we note that in other parts of the manuscript — particularly in the Discussion section (e.g., lines 359 and 373) — the wording still implies causal interpretations (e.g., "positively affected", "will reflect"). We strongly encourage the authors to revise these remaining instances to avoid suggesting causality in an observational study. Additionally, we recommend explicitly acknowledging this limitation — namely, the inability to infer causality — in the Discussion section, ideally in a standalone paragraph or as part of the study's methodological limitations.
- Literature contextualization
- Thank you for adding the citation to Carling et al. (2015). However, the sentence in the Introduction still states that “no studies have addressed managing PTL during a congested season in adult soccer players,” which remains overly broad. We recommend rephrasing this claim to acknowledge that prior work has examined training load in congested contexts, but that your study is novel in focusing specifically on pitch-based external load and its clustering in relation to match performance.
Author Response
Thanks to reviewer 2 for his important contributions in this second round of review. We welcome all suggestions as pertinent. Below are the responses point by point.
- Justification and methodological detail of cluster analysis
- Thank you for the response. The use of hierarchical clustering and dendrogram inspection provides a useful initial estimate of the number of clusters. However, including a formal validation index such as the silhouette coefficient or Davies–Bouldin index would further strengthen the argument that k = 3 is the most appropriate choice for this dataset. Even a posteriori validation (e.g., silhouette score for k = 2, 3, 4) could be insightful.
Response: We set the silhouette value for k= 2, 3 and 4. (Line 218, highlighted in yellow).
- Confounding between training load and match frequency
- Thank you for the response. However, Figure 6 is not included in the current version of the manuscript. To improve transparency, we suggest either correcting this reference or clearly illustrating in the results (possibly in a table or figure) how the adjustment for match frequency influenced the relationship between PTL cluster and MPP. Additionally, a more explicit mention of this adjustment in the Methods section (e.g., Section 2.5) would strengthen the clarity of the statistical approach.
Response: Sorry for the confusion. Figure 6 mentioned in the previous answer is actually Figure 5. We mentioned in section 2.5 (lines 233 to 235, highlighted in yellow) the statistical approach to verify the relationship between PTL clusters and MPP.
- Clarity in season segmentation (quartiles)
- Thank you for your reply and for adding explanatory content to the revised manuscript regarding the division of the dataset into quartiles. The clarification that the quartiles correspond to fixed three-month periods across both seasons helps to contextualize the temporal structure of the analysis. This was a helpful addition to the text and contributes to a better understanding of your methodological approach. That said, I would like to offer some further observations and suggestions, as a few aspects remain ambiguous or potentially confusing for readers. First, I would like to point out a small inconsistency in your response: you mentioned that the relevant clarification had been added between lines 181 and 186. However, upon reviewing the revised manuscript, the passage describing the quartile division appears in lines 193 to 200. This is a minor but important detail that should be corrected, as incorrect line references can cause unnecessary confusion during peer review and editorial processing. More substantively, although it is now stated that the numbers reported per quartile (e.g., 180, 221, 203, 117) represent “cases” and not matches, it remains unclear what exactly each “case” refers to. At various points in the manuscript, it seems that a case might correspond to a player-match observation, possibly combined with its corresponding weekly training block. However, in the revised paragraph, the wording "180 of both match physical performance and session training case observations" is ambiguous and open to interpretation. Does this mean that each case includes both a match performance observation and the associated training block? Or that these are separate counts being reported jointly?
- Given the centrality of these data to your clustering and mixed-model analyses, I strongly recommend clearly defining what is meant by a "case" early in the Methods section — ideally in section 2.1, where the sample is described. Explicitly stating that, for example, a case corresponds to a player who played a full official match and for whom the prior training data are available, would help avoid any confusion and enhance the transparency of your dataset construction. Furthermore, although your clarification attributes the unequal distribution of cases across quartiles to the higher number of matches typically played during the middle of the season, I believe this point could be made more explicitly and quantitatively. Reporting the number of matches per quartile or including a brief table or figure summarizing this information, would help demonstrate the internal consistency of the dataset — particularly given that 148 matches yielded 721 observations. This would also make it easier for readers to assess how evenly or unevenly the observational load was distributed across the season and how that may have influenced the balance of statistical comparisons. This is especially relevant for the fourth quartile, which contains significantly fewer observations (117) compared to the second (221), even though each period spans three months. It would be helpful to explain not just that this is a natural outcome of the competitive calendar, but also whether and how such sample imbalance was accounted for analytically — for instance, whether any adjustments or weighting were applied in your mixed models to account for the lower volume of data in some quartiles.
Response:
-Again, sorry for the confusion. In fact, the correction is in lines 193 to 200.
-In lines 118-121 (highlighted in yellow) we inform that data are separate counts being reported jointly.
-In lines 200-202 (highlighted in yellow) we present the number of matches per quartile.
-We used the mixed linear model using the subject ID (as random effects). This statistical option is a more appropriate approach for our data (longitudinal unbalanced data). https://rss.onlinelibrary.wiley.com/doi/epdf/10.1111/j.1467-9868.2005.00492.x
- Interpretation of causal relationships
- Thank you for making the suggested edits around lines 310–313. The language in that section has indeed been revised appropriately to reflect associations rather than causality. However, we note that in other parts of the manuscript — particularly in the Discussion section (e.g., lines 359 and 373) — the wording still implies causal interpretations (e.g., "positively affected", "will reflect"). We strongly encourage the authors to revise these remaining instances to avoid suggesting causality in an observational study. Additionally, we recommend explicitly acknowledging this limitation — namely, the inability to infer causality — in the Discussion section, ideally in a standalone paragraph or as part of the study's methodological limitations.
Response: We have revised the discussion and removed the instances related to causality.
We have added the paragraph calling attention to this limitation at lines 422 (highlighted in yellow).
- Literature contextualization
- Thank you for adding the citation to Carling et al. (2015). However, the sentence in the Introduction still states that “no studies have addressed managing PTL during a congested season in adult soccer players,” which remains overly broad. We recommend rephrasing this claim to acknowledge that prior work has examined training load in congested contexts, but that your study is novel in focusing specifically on pitch-based external load and its clustering in relation to match performance.
Response: Thank you for the observation, we have organized the sentence (lines 56, highlighted in yellow).
Round 3
Reviewer 2 Report
Comments and Suggestions for Authors
General Response to Authors
I would like to thank the authors for their careful and comprehensive revisions. The responses provided in this second round of review have addressed the main points of concern raised in my previous comments.
The additional clarification regarding the clustering methodology, particularly the inclusion of silhouette scores for k = 2, 3, and 4, strengthens the justification for the selection of k = 3. The correction of line references, improved explanation of the term “case,” and inclusion of match counts per quartile have significantly enhanced the transparency of the methodological approach. Furthermore, the explicit acknowledgment of the limitations associated with causal inference and the adjusted language in the Discussion section are appropriate and appreciated. Finally, the revised statement in the Introduction now better reflects the current literature.
Overall, the manuscript has improved in clarity and rigor, and I appreciate the authors’ responsiveness and effort in refining their work.
Kind regards,
Reviewer 2